# Encoding and Decoding of p53 Dynamics in Cellular Response to Stresses

**DOI:** 10.3390/cells12030490

**Published:** 2023-02-02

**Authors:** Ping Wang, Hang-Yu Wang, Xing-Jie Gao, Hua-Xia Zhu, Xiao-Peng Zhang, Feng Liu, Wei Wang

**Affiliations:** 1Kuang Yaming Honors School, Nanjing University, Nanjing 210023, China; 2Key Laboratory of High Performance Scientific Computation, School of Science, Xihua University, Chengdu 610039, China; 3Institute of Brain Sciences, Nanjing University, Nanjing 210093, China; 4National Laboratory of Solid State Microstructure, Nanjing University, Nanjing 210093, China; 5Department of Physics, Nanjing University, Nanjing 210093, China

**Keywords:** p53 dynamics, encoding, decoding, network model, cell fate

## Abstract

In the cellular response to stresses, the tumor suppressor p53 is activated to maintain genomic integrity and fidelity. As a transcription factor, p53 exhibits rich dynamics to allow for discrimination of the type and intensity of stresses and to direct the selective activation of target genes involved in different processes including cell cycle arrest and apoptosis. In this review, we focused on how stresses are encoded into p53 dynamics and how the dynamics are decoded into cellular outcomes. Theoretical modeling may provide a global view of signaling in the p53 network by coupling the encoding and decoding processes. We discussed the significance of modeling in revealing the mechanisms of the transition between p53 dynamic modes. Moreover, we shed light on the crosstalk between the p53 network and other signaling networks. This review may advance the understanding of operating principles of the p53 signaling network comprehensively and provide insights into p53 dynamics-based cancer therapy.

## 1. Introduction

Cells exhibit powerful information processing capabilities when they are exposed to various stresses such as DNA damage, hypoxia, and nutrition deficiencies [1,2]. In-depth studies of how cells respond to stresses are helpful for defeating malignant diseases including various cancers. A cell can accomplish information processing through various signaling networks. Usually, there is a specific hub that can be regarded as an information processing center, which identifies and integrates the upstream signals and further activates the downstream effectors to guide cellular outcome. It is strongly believed that the orderly regulation of cellular signaling is responsible for maintaining the normal tissue functions and that its dysfunction may result in various diseases. However, given the technical limitations and complex intracellular regulation, deciphering how cells autonomously perform the entire process of signal transduction is still a challenging task.

Transcription factors (TFs) are proteins that can bind to the promoters of target genes specifically to regulate their expression. Due to their versatility, TFs are usually considered as information processing centers. As one of the most important tumor suppressors, p53 mainly acts as a transcription factor to maintain genomic integrity by inducing a large number of target genes in cellular response to stresses [3]. In unstressed cells, p53 is kept at low levels due to its rapid degradation. In response to multiple stresses, p53 accumulates in the nucleus to modulate multiple processes such as cell cycle arrest, apoptosis, and senescence. The experimental observation of p53 dynamics in single cells was largely advanced by the time-lapse microscopy and fluorescent labeling techniques in living cells [4]. Different patterns of p53 dynamics have been observed including pulses, sustained platforms, monotonic increasing and two-phase dynamics (pulses followed by sustained high levels) [4,5,6,7]. The emergence of these dynamics is shown to be related to the cell type and the intensity of the applied agents. A negative feedback loop consisting of p53 and its negative regulator Mdm2 has been identified as the core topology in regulating p53 dynamics. It is less clear how p53-Mdm2 loop is differentially modulated to shape p53 dynamics in cellular response to various stresses and how the resulting dynamics selectively activate the target genes.

A fundamental strategy in understanding the underlying mechanisms of various dynamics is to seek out the corresponding regulatory networks in response to specific stresses. Based on the given functions of the network motifs, we can make some speculations about which kind of motif is responsible for the observed dynamics. The alternate activation of p53 and Mdm2 is responsible for p53 oscillations upon ionizing radiation (IR) [4]. Differences in the response of p53 to IR and UV radiation reveal the significance of the p53-Wip1-ATM negative feedback loop in generating p53 oscillations [8]. On the other hand, it is possible to probe the differences in the strength of a specific interaction in the signaling network by comparing p53 response to the same damaging agent in different cell lines. It is the strong inhibition of Mdm2 by ATM that makes U-2 OS cells more sensitive to etoposide, thus producing the bimodal p53 dynamics in these cells [6,7]. It seems unrealistic to experimentally track multiple nodes simultaneously and record the entire process of signal transduction. Combining the experimental approaches with the mathematical modeling is a feasible means to identify the activated subnetworks in p53 response to a specific stress. A valid model can provide predictions that point out a possible direction of efforts for further experiments and can further establish iteration with the experiments. A good example of modeling on p53 network is the earlier work proposed the two-phase dynamics of p53 in the DNA damage response, which was successfully validated in doxorubicin-treated MCF7 cells [9,10].

In this review, we discuss recent advances in understanding the underlying mechanisms in the generation of p53 dynamics and the decision of cell fates in response to stresses. We summarize the latest findings in two aspects: encoding—how DNA damage is encoded into p53 dynamics and decoding—how cell fate is decoded from p53 dynamics. To give an overview of p53 response to DNA damage, we summarize the p53 signaling network including the damage types, sensors, upstream circuits, dynamics, transcription, and downstream circuits associated with different cell fates (Figure 1). Moreover, at the end of each part, we review the relevant progress in modeling. Finally, we illustrate the contribution of modeling to understanding dynamic transitions of p53 and giving insights into the interplay between p53 networks and other signaling networks.

## 2. Encoding: Translating DNA Damage into p53 Dynamics

### 2.1. Generation, Sensing, and Repair of DNA Damage

The integrity of the genome is challenged by genotoxic stresses. Failure in cellular response to such stresses is closely related to carcinogenesis. During the cell cycle, cells are subject to transient and intrinsic DNA damage during mitosis. Extrinsic DNA damage agents mainly include antitumor drugs (e.g., Dox (Doxorubicin), etoposide, cisplatin, and NCS (Neocarzinostatin)) and irradiations. These damage agents can produce single- or double-strand breaks (SSBs or DSBs) in DNA. As the input to the p53 signaling network, different types of DNA damage activates the downstream pathways specifically, leading to different cellular outcomes (Figure 1).

SSBs are breaks in one strand of the DNA double helix, mainly caused by UV radiation. They can be resolved by homologous recombination (HR) and alternative homologue-mediated SSB repair pathways [11]. DSBs are the predominant DNA lesions caused by IR, Dox, and etoposide. Two distinct pathways exist for the repair of DSBs: nonhomologous end joining (NHEJ) and homologous recombination (HR) [12,13]. HR mainly contributes to DSB repair in S phase, whereas NHEJ is predominantly activated in G1 phase [14]. In modeling, the Monte Carlo method can be used to simulate the DNA repair dynamics in the case of DSBs [15]. In each cell, the initial number of DSBs is assumed to follow Poisson distribution. In addition, three states are considered in the DSB repair process: intact DSB, DSB–protein complex (DSBC) and fixed DSB [9,16]. Failure in DNA repair may lead to a sustained p53 response like the persistent oscillations observed in MCF7 cells [17].

Cells sense DNA damage by activating several kinases including ATM, ATR, and DNA-PKcs, which are accountable for detecting damage lesions via the MRN (MRE11-RAD50-NBS1) complex and transmitting damage signals to the tumor suppressor p53. Upon DNA damage, ATM is specially activated by DSBs through phosphorylation, regulating DSBs repair and activating p53 [18]. Based on its activity, ATM can be divided into three forms: inactive dimers, inactive monomers, and active monomers [18,19]. It has been reported that low levels of DSBs are sufficient to activate ATM [20]. In contrast, ATR is an important detector of SSBs. ATR can also implement self-activation by phosphorylation [21]. Both ATM and ATR can activate p53 by phosphorylating it [1]. Moreover, ATM-dependent phosphorylation of Mdm2 can accelerate its degradation, thereby promoting p53 activation indirectly [22,23], while ATR-dependent Mdm2 phosphorylation only inhibits the activity of Mdm2 for degrading p53 [24]. DNA-PKcs can be activated by both DSBs and SSBs, stabilizing p53 by phosphorylating both p53 and Mdm2 [25,26,27,28]. In particular, the main phosphorylation sites involved in the above processes are marked in Figure 1.

ATM, ATR, and DNA-PKcs are all members of phosphatidylinositol 3-kinase-related kinases (PIKKs) family. Although they can sense DNA damage independently, there is a crosstalk between them in some cases [29]. DNA-PKcs can also be phosphorylated by ATM [30] and ATR [31]. DNA-PKcs collaborates with ATM to complete the response to DNA damage [25]. ATM is hyperactivated when the catalytic activity of DNA-PKcs is blocked, hinting that DNA-PKcs may suppress ATM activity [29]. Interestingly, all three sensors can be activated in response to IR, and interfering with their interaction can significantly affect the sensing of DNA damage, thereby modulating the dynamics of p53. The accumulation of unrepaired DSBs due to the inhibition of DNA-PKcs may signal to ATM and activate p53, leading to the enlarged first pulse of p53 [32].

### 2.2. Stimulus-, Cell Type- and Species-Dependent p53 Dynamics

Upon DNA damage, p53 levels vary over time and exhibit rich dynamics (Figure 2). Lahav et al. obtained individual live-cell data on p53 expression by time-lapse imaging [4]. They found that p53 shows digital pulses in which the number of pulses, rather than the size, increases with the doses of IR on average. Meanwhile, their results may advance the understanding of the link between p53 digital pulses in single cells and the damped oscillation at the population level [33]. Furthermore, the p53-Mdm2 negative feedback loop is responsible for this novel dynamics and functions as a “digital” clock until the damage is repaired [4]. With deficiency in DNA repair, it was observed that p53 pulses last for three days in MCF7 cells [17]. Digital pulses indicate repeated detection of DNA damage and imply a return to normal proliferation after DNA repair, whereas persistent oscillations can induce irreversible cellular outcome. In addition, spontaneous p53 pulses can be triggered by an excitable mechanism when mitotic damage appears in the normal cell cycle [25].

In response to UV-induced DNA damage, a single extended pulse of p53 is induced and its width and height increase with the UV dose [5] (Figure 2b). The absence of persistent p53 pulses in response to UV radiation results from the inactivation of the ATM-p53-Wip1 feedback loop [5]. p53 triggers apoptosis in response to UV in contrast to cell cycle arrest induced by IR. It is plausible that p53 dynamics should be associated tightly with cellular outcomes in the DNA damage response. Lahav et al. artificially altered p53 dynamics from pulses to a platform with a height equaling the pulses by treating the cells with an inhibitor of Mdm2, Nutlin-3 [34]. As a result, sustained p53 expression induces cellular senescence instead of apoptosis. The above results validate the significance of p53 dynamics in determining cell fate, suggesting that a much higher p53 concentration may be required for apoptosis induction.

Notably, the above results were observed in MCF7 cells, which were later found to be insensitive to a spectrum of antitumor drugs including etoposide, Nutlin-3a, and 5-Fluorouracil [35]. Indeed, the bimodal switching of p53 dynamics is observed in etoposide-sensitive U-2 OS and A549 cell lines with increasing DNA damage: p53 shows low-amplitude oscillations and monotonic increasing upon mild and severe damage, respectively [6] (Figure 2e,f). When HCT116 cells are treated with cisplatin, there exists remarkable variability between individual cells in apoptosis induction depending on the temporal variation of p53 [36]. Moreover, Wu et al. found that p53 shows two-phase dynamics in Dox-treated MCF7 cells, i.e., oscillations followed by a high terminal pulse, and the duration for oscillations shorten with increasing doses of Dox, meaning accelerated apoptosis induction [10] (Figure 2a). The two-phase dynamics may result from the sequential transition in the domination of distinct feedback loops [9]. In contrast to the sharp increasing that induces immediate apoptosis, these two-phase dynamics containing several pulses correspond to the attempt for DNA repair before apoptosis induction.

There exists another mode of two-phase dynamics of p53: cells exhibit a high-amplitude p53 pulse in the first phase and undergo low-amplitude p53 oscillations in the second phase [37]. This kind of p53 dynamics can occur by depleting MdmX alone, another negative regulator of p53. Of note, MdmX deletion promotes or represses the subsequent UV-induced apoptosis depending on the interval between the two treatments. The first high pulse always coincides with mitosis, suggesting that the mitotic damage and deletion of MdmX cooperate to amplify the accumulation of p53 [37]. However, the above mechanism for p53 dynamics is not suitable for the similar two-phase time series of p53 observed in 769-P and HepG2 cells treated with etoposide [7]. In that case, it is the weaker inhibition of Mdm2 by ATM that prevents p53 from marked accumulation in the second phase, thus allowing p53 to return to low oscillations and desensitizing these two types of cells. Accordingly, combined inhibition of Mdm2 and Wip1 was thought to be an effective strategy to increase p53 expression and induce the desired apoptotic response [7]. Together, comparative studies between different cell lines can easily identify the disadvantages of nonsensitive cell lines and find a clue for killing the corresponding cancer cells.

Different p53 dynamics in various cell lines have inspired the investigation into whether such differences in p53 dynamics exist in different tissues or species. Lahav and coworkers observed that the radiosensitivity corresponding to p53 dynamics shows remarkable differences between tissues. The small and large intestines are relatively insensitive to p53 levels peaking at 2–3 h followed by decreasing in its levels, while lymphoid organs, including the spleen and pancreas, are more sensitive to radiation with sustained p53 expression [38]. In another independent work, they observed consistency and variability in p53 oscillation dynamics across species [39]. Although p53 oscillations appear in different species in response to IR, p53 oscillates with higher frequency in rodents than in humans and other species. Furthermore, the core feedback model based on p53-Mdm2 feedback loop revealed that stronger negative feedback between p53 and Mdm2 should be the driving force for faster oscillations in rodents. Indeed, the temporal dynamics of each species may be conservative for evolution. In humans, the oscillation period is about 5.5 h on average [17]. The evolution of a new p53 pulsing frequency would be an undesirable event as it would require simultaneous changes in the expression dynamics of a series of p53 target genes [40].

### 2.3. Clarifying the Mechanism in the Encoding of p53 Dynamics by Modeling

The mathematical modeling on p53 network provides a theoretical perspective to understand the generation mechanism of p53 dynamics. The negative feedback loop between p53 and Mdm2 governs p53 oscillations upon DNA damage [4]. Nevertheless, a two-variable model composed of only p53 and Mdm2 is not sufficient to generate persistent p53 oscillations; a sufficient explicit delay or an implicit delay introduced by a long negative feedback loop or additional positive feedback loop is required for producing oscillations [41]. It also enlightens us that the absence of sustained oscillations in MCF7 cells exposed to UV may result from the insufficient ‘nonlinearity’ and imbalance in the p53-Mdm2 feedback loop [5,41].

The core status of the p53-Mdm2 loop in p53 regulation is challenged by the connection of p53 pulses to the pulses of the damage sensor ATM [8]. p53-induced Wip1 was shown to inactivate ATM by dephosphorylation, thereby deactivating p53 and enclosing a negative feedback loop. Inhibiting Wip1 expression greatly reduces the amplitude of p53 pulses in the irradiated MCF7 cells and impairs the regular oscillations, highlighting the significance of recurrent initiation of upstream sensor in producing persistent pulses of p53 [8]. The simulation results showed that the ATM-p53-Wip1 loop with an intrinsic time delay is important for the generation of p53 pulses [9,42]. When the ATM-p53-Wip1 loop is shut off, p53 level exhibits damped oscillations. Moreover, Wip1 also controls the stepwise activation of p53 and plays a significant role in p53-controlled cell fate decision [16,19]. However, the requirement of the ATM-p53-Wip1 loop in p53 oscillations seems to be related to the type of stresses. In the case of p53 oscillations generated by deleting MdmX alone, ATM was not activated in the absence of DNA damage [37]. Instead, inhibition of Mdm2 resulted in nonoscillatory dynamics of p53, confirming the crucial role of p53-Mdm2 feedback loop in shaping p53 oscillations.

Long-term measurements revealed that p53 oscillations exhibited marked heterogeneity, especially in the oscillation amplitudes [17]. Based on the mathematical modeling, it was shown that the low-frequency noise in the protein production rate should be the source of variability in the oscillations [17]. By introducing stochasticity in the DNA repair module, a network model including the p53-Mdm2 loop is suggested to explain the cell-to-cell variation in p53 oscillations [43]. This study reveals that upstream signaling can greatly influence the downstream p53 dynamics. Thus, it is not difficult to understand that the distinct dynamics of p53 in different cell lines are related to the efficiency of DNA repair and the activity of kinase ATM [44].

The generation of basal dynamics of p53 depends heavily on an excitable mechanism. A hallmark of an excitable system is that a transient input is sufficient for triggering a full response. Thus, similar to the case of cellular response to extrinsic damage, a complete p53 pulse will be initiated by transient damage during mitosis [25]. Batchelor et al. found that p53 network is just an excitable system [5]. Compared to UV radiation, IR is more likely to activate the excitability of the p53 network due to ATM-induced rapid degradation of Mdm2 [5,8]. Therefore, p53-Mdm2 and p53-Wip1-ATM feedback loops may cooperate in controlling the excitability of p53 [5,8,43]. The p53-Mdm2 loop should be the topology basis for excitability in p53 dynamics, while the p53-Wip1-ATM loop modulates sensitivity of the excitable response of p53 [43]. Based on the coupling of these two loops, Sun et al. supposed that daughter cells may inherit considerable amount of phosphorylated ATM, leading to a p53 pulse through excitable mechanism. Furthermore, Chong and coworkers paid more attention to p53 autoregulation, which modulates the threshold of excitability [45].

The coupling of two negative feedback loops is important for producing some specific properties of the p53 system, such as excitability (Figure 1). It has been proposed that a single negative feedback loop can induce oscillation response while it is not sufficient to maintain a uniform and sustained oscillation [46]. Thus, in the p53 system, an additional negative feedback loop like p53-Wip1-ATM loop is required to tune the oscillations produced by the primary p53-Mdm2 loop [5,8,9]. Possibly, this additional loop can make p53 oscillation robust to noise because there may exist a wider range of parameters in producing oscillations in the case of coupled feedback loops than the case of a single negative feedback loop [46]. Moreover, coupling negative feedback loops can further accelerate the response and maintain the stability of the system. How p53 system employs the coupling of two negative feedback loops to induce more intriguing dynamics requires further investigation.

Interconnected positive and negative feedback loops (IPNFLs) are another prevalent coupling pattern in producing p53 pulses [47] (Figure 1). The two types of loops contain their own unique characteristics, thus allowing p53 to exhibit tunable dynamics. Compared to a single positive feedback loop, IPNFLs have features of both noise represser and response accelerator. On the other hand, due to the modulation of negative feedback by positive feedback, the system including IPNFLs achieves a widely tunable frequency and near-constant amplitude [48]. However, the unbalanced combination of two distinct feedback loops sometimes drives this coupled system to show either oscillation or bistability [49]. If the negative feedback loop is much stronger than the positive one, the system shows oscillations, and vice versa, there exists bistability. For p53 network, it is the temporal alternation in the predomination of the two feedback loops leads to the two-phase dynamics of p53 [9]. Coupling of p53-Mdm2 negative feedback loop with p53-PTEN-Akt-Mdm2 positive feedback loop facilitates the switching of p53 dynamics from oscillation to bistability when the positive feedback loop is dominant in the late phase of cellular response to severe damage [9]. The positive feedback loop including ATM autoregulation also contributes to driving excitable p53 dynamics [43]. Besides PTEN, miR-605 is a direct transcriptional target of p53, which can form p53-Mdm2-miR-605 positive feedback loop by repressing the expression of *mdm2*. This positive feedback loop may control the amplitude of p53 pulses [50].

In response to DNA damage, p53 is gradually activated to perform different functions and different feedback loops modulate p53 dynamics in this process. In modeling, it was assumed that there exist different forms of p53 based on its phosphorylation status [9,16,19,47]. In response to low-dose of IR, p53 is activated to form “p53 arrester” that participates in the p53-Mdm2 feedback loop and induces cell cycle arrest. For high-dose of IR, p53 is further activated to form “p53 killer” in the late phase of the response that will kill the damaged cells by inducing apoptosis. The p53-PTEN-Akt-Mdm2 positive feedback loop contributes to the further amplification of p53 in this form. Under UV radiation, HIPK2 forms a double-negative feedback loop with Mdm2 to accelerate the conversion of p53 arrester to p53 killer [51]. This conversion can also be facilitated by DYRK2, which is in cytoplasma in resting cells but translocates to the nucleus upon DNA damage [52]. Programmed cell death 5 (PDCD5) is also found to enhance p53 killer formation by dissociating the p53-Mdm2 complex and promoting Mdm2 degradation [53].

Moreover, other functional factors also play some roles in the encoding of p53 dynamics. Multiple miRNAs, including miR-605, miR-192, miR-29a, and miR-34a can directly or indirectly repress Mdm2, thus forming a p53-miRNA-Mdm2 positive feedback loop [54,55]. These miRNAs may enhance the robustness of p53 oscillations [55] and promote cell survival [54]. p300, HDAC1, and p14ARF are involved in the regulation of the p53-Mdm2 loop and thus modulate the oscillatory behavior of p53 [56,57]. Circadian factor Period 2 (Per2) prevents the degradation of p53 by Mdm2 to dictate the phase of p53 oscillations [58]. Together, the dynamics of p53 largely depend on the properties of the circuits. More studies on p53 dynamic properties require both experimental and modeling investigation in future.

## 3. Decoding: Controlling Cell Fate by p53 Dynamics

### 3.1. p53 Dynamics-Dependent Selection of Target Genes

In response to stresses, p53 induces hundreds of target genes to modulate a wide range of cellular processes including cell cycle arrest, apoptosis, senescence, DNA repair, and metabolism [3,59,60]. This huge cohort of target genes controlled by one inducer raises a question: how does p53 selectively activate its target genes performing specific functions to respond to various stresses? p53 exhibits rich dynamics in response to different stresses which reminds us that p53 dynamics could be a key determinant of this selectivity. p53 “abundance” in the mono-pulse upon UV or monotonic increasing upon etoposide treatment is much higher than the amplitude of p53 oscillations [5,6]. The abundance could be a significant factor in cell fate decision since low p53 expression predominantly induces genes like p21 to induce cell cycle arrest while high p53 expression tends to activate genes like BAX to trigger apoptosis [61]. The molecular mechanism supporting this view lies in an “affinity model”, in which p53 has a much lower affinity for the promoters of proapoptotic genes than for those of proarrest genes, thereby making a threshold-dependent cell fate decision between growth arrest and apoptosis [62]. Artificial interference of Mdm2-dependent p53 degradation shifts oscillations into sustained p53 expression equal to the peak of the oscillations, transforming cellular outcome from transient cell cycle arrest to senescence [34]. However, subsequent study reported that both surviving and apoptotic cells treated by antitumor drugs reach similar levels of p53 and the p53 threshold required to enact apoptosis rises with time, suggesting that additional factors are involved in the decision between survival and death [36].

Over the past decade, much effort has been put into exploring the selective expression of the target genes by oscillatory p53. By experimental measurements, the amplitudes of p53 pulses were found to delineate promoter activation thresholds, while pulse frequencies were differentially filtered by the promoters of the target genes [63]. These results suggest that the promoter of mdm2 gene is more sensitive to p53 oscillations, thereby making the spontaneous p53 pulse caused by transient damage sufficient to activate Mdm2 instead of p21 [25]. Thus, it gives us insight that the physical properties of p53 oscillations, such as amplitude or frequency, can help distinguish circuit-regulation genes (mdm2) from cell cycle-regulation ones (p21).

Under p53 oscillations, p21 mRNA tightly follows p53 pulses, while p21 protein has three different patterns: oscillation, slow, and rapid accumulation [64]. The differential accumulation of p21 results from different degradation rates of the protein instead of DNA binding dynamics of p53. Moreover, the proteins of multiple p53 target genes exhibit distinct dynamic patterns when the target genes show similar pulsatile DNA binding of p53, suggesting the involvement of other post-transcriptional mechanisms in regulating the protein dynamics [65]. From a transcription perspective, the expression level of p53 mainly increases the probability of the transcription of its target genes and the transcription magnitude of the targets can saturate easily due to increasing p53 [64]. Follow-up studies revealed that the difference between the expression dynamics of p53 target mRNAs and the corresponding proteins may result from the difference in the degradation rate of the mRNAs and proteins [66]. Quantification of the central dogma in the p53 pathway revealed that the persistent accumulation of p21 protein levels over time resulted from a shorter mRNA half-life and a longer protein half-life of p21 compared to the duration of p53 pulses [64]. A four-quadrant classification of mRNA and protein decay rates compared to p53 pulsing frequency showed that the genes involved in cell cycle arrest are the highest fold enrichment in the first quadrant that holds pulsing dynamics, while genes involved in apoptosis showed enrichment in the second quadrant that shows as pulse counter [66]. These results strongly suggest that different mRNA and protein dynamics may separate the targets into different functional classes in the presence of p53 oscillations. However, there is still a long way to generalize this notion, as the authors only considered 36 well-characterized p53 targets and we also need to think about why some proapoptotic genes would inappropriately appear in the first quadrant [66]. Actually, in a study concerning an extended number of targets involved in more biological processes, it was shown that the expressions of some genes can be both pulsing and monotonic rising in the time courses [40].

Additional mechanisms, including different activation thresholds and the presence of feedforward loops, should contribute to modulating the exclusive induction of targets under specific p53 dynamics [67]. Meanwhile, oscillatory p53 dynamics were recognized to have a higher capacity to diversify target gene expression profiles than the sustained expression. Since the expression of target genes is closely related to cellular functions, a possible hypothesis is that p53 sustained expression is responsible for terminating cell fates, such as apoptosis or senescence, while p53 oscillations are implicated in regulating a variety of transient cellular outcomes, such as cell cycle arrest and metabolism. Identifying how p53 oscillations distinguish these pro-survival processes requires further exploration.

### 3.2. Multilevel Modulation of p53 Targets Selection

In addition to the dynamic patterns of p53, there are multiple levels of regulation that can help p53 select targets to execute appropriate cellular responses [68]. Upon DNA damage, p53 shows rapid tetramerization from its dimeric form and is able to bind dynamically to the target promoters [69]. As a result, the bound fraction of p53 to the promoter increases, and acetylation of the p53 CTD (C-Terminal Domain), particularly at K382, could significantly prolong its residence time and thus enhance the transcriptional activation. Consistently, transcription of the targets such as p21 shows dependence on the acetylation status of p53 [25]. Indeed, the length of residence time can be regarded as another interpretation of the affinity of p53 to the promoters of its targets. An increase in promoter activity results in an enhancement in the burst frequency or burst size. It was found that burst frequency can be used to distinguish different targets: Mdm2 and p21 belong to the “transient” group, while BAX and DDB2 belong to the “sustained” group [70]. Thus, p53 can selectively induce its target genes by specifically modulating its own transcriptional activity at the promoters [71].

The acetylation status assists p53 in preferentially activating a specific cell fate, such as apoptosis, highlighting the importance of post-translational modifications in regulating the target selectivity of p53 (Figure 1). Gu et al. found that Tip60-dependent p53 acetylation at K120 is essential for apoptosis instead of growth arrest [72]. This acetylation process is GSK-3-dependent, as activation of Tip60 requires GSK3-mediated phosphorylation and the resulting acetylated p53 can induce PUMA to initiate apoptosis [73]. In connection with phosphorylations that can stabilize p53, we speculate that the multiple post-translational modifications on p53 are a stepwise process in the modeling work [42]. That process depends on the intensity of damage and correlates with the selection of target genes: mild damage causes initial phosphorylation of p53 on Ser15/20, which stabilizes p53 to induce Mdm2-dependent feedback regulation and p21-mediated cycle arrest, while severe damage promotes further phosphorylation of p53 on Ser46 and acetylation on K120, elevating p53 and activating the proapoptotic targets. The specific acetylation on K120 appears only in highly expressed p53, suggesting that post-translational modifications can accompany with specific dynamic pattern to determine cellular outcome [73]. The main types of post-translational modifications on p53 include SUMOylation, neddylation, phosphorylation, acetylation, methylation, ubiquitination, hydroxylation, O-GlcNAcylation, ADP-ribosylation, and β-hydroxybutyrylation [74]. These modifications across more than 36 sites on the p53 peptides suggest that a way to decide cell fate may be embedded in the systematic regulation of the post-translational modifications of p53 [75,76].

### 3.3. Decoding p53 Dynamics into Cell Fate through the Downstream Circuits

The selective induction of p53 targets between cell cycle arrest, senescence, and apoptosis has been extensively explored. p53 leads to cell cycle arrest by inducing p21 [77,78]. The ability of p53 to induce cell cycle arrest is also required for senescence. When cell cycle arrest becomes irreversible, cells enter a state of senescence. p21, pRB, and E2F family proteins play significant roles in p53-induced senescence [79] (Figure 1). Apoptosis is another kind of terminal cell fate. A large number of apoptotic genes are targets of p53 and are involved in various steps of apoptosis signaling and execution. The common proapoptotic targets of p53 include the proapoptotic proteins (PUMA, Bad, BAX, Bak) in Bcl-2 family, death receptors (Fas, Dr4, Killer/Dr5), and downstream apoptotic factors (Apaf1, p53AIP1, caspase 6, caspase 3 (Casp3)) (Figure 1). Induction of the BH3-only proteins by p53 causes mitochondrial outer membrane permeabilization (MOMP) [80]. The induction of the proapoptotic targets of p53 results in the release of various cell death modulators from the mitochondria, such as cytochrome c (Cytoc). Then in the cytosol, Cytoc engages Apaf1 to form the apoptosome, activating caspase 9 (Casp9) [81]. These are key steps in the intrinsic apoptosis pathway. The factors mentioned above are responsible for decoding p53 dynamics in modeling and are closely related to the final cell fate.

A great deal of modeling work has focused on the choice between survival and apoptosis in cellular response to stresses. In general, differential activation of BAX, PUMA, Apaf1, Casp3, and p21 can be exploited to indicate different cellular outcomes in the models [47,82]. Based on the “affinity model”, their activation depends on the “abundance” of p53 with different dynamics, including oscillations and sustained rising. It was assumed that low oscillatory p53 is sufficient to induce p21 to trigger cell cycle arrest, while sustained high levels of p53 prefer to transactivate proapoptotic factors, including PUMA [83], BAX [52,84], and Apaf-1 [9,16,51,52]. A significant characteristic of apoptosis is its irreversibility. Positive feedback loops are widely used to ensure irreversible activation of caspases. For example, positive feedback loop between Cytoc release and Casp3 activation is considered to realize irreversibility in the activation of Casp3 in modeling [53,85]. Cytoc is activated to induce Casp3 when its expression exceeds a certain threshold. Then, Casp3 is fully activated and is only restored when the upstream stimulus level is lower than a sufficiently small threshold. As an upstream stimulus, p53AIP1 can activate Casp3 activation irrepressibly if the basal level of p53AIP1 can maintain the activation of Cytoc or Casp3. Thus, the apoptotic switch becomes irreversible once it is turned on. That is consistent with the previous experimental observation that apoptosis is really irreversible after “the point of no return” [16,19].

The regulation of apoptosis becomes complex when multiple factors are involved. Activated E2F-1 induces ASPP, which promotes the expression of BAX to trigger apoptosis [83]. p53-indued miR-22 represses p21 expression and activates E2F-1, which was also shown to lead to apoptosis [54]. miR-34a promotes p53-dependent apoptosis by suppressing the expression of antiapoptotic genes such as Bcl-2 [50]. PDCD5 helps regulate BAX translocation in the cytoplasm and enhances the Cytoc-Casp3 feedback loop [53]. It is meaningful to investigate how p53 distinguishes a status of senescence from apoptosis. Although the regulation of senescence is less well understood, it should be tightly associated with activation of the p53/p21${}^{\mathrm{cip}}$ pathway and the p16${}^{\mathrm{INK}4a}$/pRB pathway [86]. A Boolean model is developed to investigate cell-fate decisions contemplating three possible phenotypes: autophagy, apoptosis, and senescence [87]. Similar to the previous models, p53 arrester is employed to induce p21 to promote senescence. In contrast, p53 killer activates DNA damage-regulated autophagy regulator 1 (DRAM1), which induces the Unc-51-like kinase 1 (ULK1) protein complex specifically for autophagy induction. This system would exert apoptotic effects when Casp3 and DRAM1 are activated by the p53 killer while ULK1 is inactive.

## 4. Mathematical Modeling Provides Insights into a Comprehensive Understanding of p53 Response by Integrating the Encoding and Decoding Processes

### 4.1. Understanding the Transition between p53 Dynamic Modes

p53 shows persistent pulses at low etoposide doses but exhibits monotonic increasing at rather high doses in U-2 OS cells [6]. It is intriguing to reveal how p53 behaves at moderate doses of etoposide, and how p53 oscillations switch to sustained high expression upon severe damage. Given the respective characteristics of these two types of dynamics, it is speculated that the transitional dynamics should contain both features of oscillations and rapid rising. The earlier modeling work proposed two-phase dynamics of p53 in which several p53 pulses followed by rising to high levels in response to severe damage [9] (Figure 2a). Although these transitional dynamics may be not applicable to etoposide-treated U-2 OS cells [35], similar pattern of p53 dynamics was observed in Dox-treated MCF7 cells [10]. The number of pulses in the transient oscillatory state was found to be variable [10] and may decrease with agent level, indicating a variable rate of apoptosis induction. The mechanism of the above dynamics may lie in the gradual alternation in the predomination from a negative feedback loop like p53-Mdm2 loop to a positive feedback loop like p53-PTEN-Akt-Mdm2 loop, leading to the transition from oscillation to sustained p53 expression at high levels. Meanwhile, p53DINP1 may act as an integrator to facilitate a gradual transition from negative to positive feedback loops, thus allowing the existence of a long period of transient pulses while pushing p53 to a high expression when crossing a threshold [16]. Consistently, the activation of the positive feedback loop is much later than that of the negative one, and this is required for the presence of a low transient oscillation state before switching to high sustained levels [49]. In addition, if positive feedback is activated rapidly, accompanied by appropriate parameters and initial conditions, there will be an “early switching”, a phenomenon characterized by the value of the driven parameter required for jumping to a high state smaller than the threshold corresponding to the saddle-node bifurcation [49]. Identifying the transition dynamics and clarifying the underlying mechanisms allows us to gain a global understanding of the response of p53 to different intensities of stresses. Disruption or augmentation of key nodes that control dynamic transitions may give rise to a favorable pattern of p53 response in cancer therapy.

It is also important to investigate how p53 oscillations terminate after DNA repair. Recently, it has been proposed that p53 undergoes an intermediate state between early high-frequency oscillations and late low-frequency pulses and finally returns to a basal state [43] (Figure 2c). It can be speculated that the time required for DNA repair determines the number of pulses in transient oscillations before returning to the basal state [43]. Therefore, p53 network can perform different functions including repair and apoptosis depending on the severity of DNA damage. When the cells can repair the damage efficiently, they recover to normal proliferation after DNA repair; while excessive damage is beyond the capability of repair, apoptosis is induced to kill the damaged cells.

It was recently reported that the transition dynamics of p53 from oscillations to high rising in etoposide-treated U-2 OS and A549 cells may sequentially undergo damped oscillations and a rising with a moderate platform [35] (Figure 2f). The transition from disruption of regular oscillations to damped ones suggests a regulatory effect of p53-Wip1-Mdm2 feedback loop on the p53-Mdm2 pathway. The disruption of the interaction between Mdm2 and p53 increases the stability of p53 and is responsible for the increasing of p53 to different steady states. By contrast, p53 transits from oscillations to a single expanded pulse in etoposide-insensitive MCF7 cells. Thus, the significant difference between sensitive and insensitive systems is the terminal dynamics instead of the transition dynamics. Notably, in Nutin-3 and 5-FU-treated cells, p53 oscillations are difficult to maintain even at low doses [35]. A careful study of the mechanism of p53 dynamics under different agents revealed that the stimulants act directly on the p53-Mdm2 loop, such as Nutlin-3, which interferes with the binding of Mdm2 and p53, inducing a rapid increase in p53 after disrupting its oscillation. If the stimulus acts indirectly on the p53-Mdm2 loop, it is easier to observe p53 oscillations in the transition dynamics [9,16]. Furthermore, in Nutin-3 and 5-FU-treated U-2 OS cells, the transition dynamics of p53 were found to increase at different rates and may converge to a fixed steady-state [35] (Figure 2e). The rate of increase was shown to depend on the strength of the additional positive feedback loop and may alter the apoptosis threshold similar to the case in cisplatin-treated HCT116 cells [36].

The study of transition dynamics suggests that we can look for clues to kill cancer cells by amplifying p53 signaling. We can choose an agent that is sensitive to the specific cells. For example, MCF7 cells are sensitive to UV light but not to NCS and Etoposide [5] (Figure 2b–d). This example also instructs us that NCS-induced ATM is mainly used to maintain p53 oscillations, while UV-induced ATR can amplify the p53 signal. Without accounting for side effects, Nutlin-3 may represent an ideal class of agents that act directly on Mdm2 to activate p53, more readily allowing it to reach higher levels to promote apoptosis in cancer cells [88]. In fact, the inhibition of MdmX, another inhibitor of p53, was not as effective as that of Mdm2. Deletion of MdmX alone does not induce apoptosis and should be combined with UV irradiation to kill cancer cells [37].

### 4.2. Coupling p53 Network with Other Signaling Networks in Cell Fate Decisions

The p53 network has a key role in response to cellular stresses, creating scenarios in which the p53 network couples with other signaling networks. The establishment of a coupling relationship is supported by several aspects: (I) coactivation by the same stimulus (e.g., DNA damage or hypoxia); (II) sharing some key regulatory factors (e.g., p300); and (III) coregulation of the same cellular process. The coactivation of multiple signaling networks is the basis of network coupling, and the abundance of coregulators determines the degree of activation of individual networks, thus affecting the dynamic properties and functions of each network (Figure 3).

Hypoxia can activate both p53 and HIF-1, thus causing an interplay between these two transcription factors. However, their activation is different and determined by the severity of hypoxia: HIF-1 is activated to promote glycolysis and angiogenesis upon mild and moderate hypoxia, while only upon severe hypoxia or anoxia, p53 is induced to trigger apoptosis [42]. Our modeling study revealed that active HIF-1 has the ability to promote the accumulation of p53 phosphorylation through transcription upregulation of PNUTS, thus causing an initial interaction between these two transcription factors upon severe hypoxia [42,89,90]. Upon severe hypoxia, partially activated p53 represses HIF-1 indirectly by competing with HIF-1 for the coactivator p300 [91]. Upon anoxia, fully activated p53 promotes HIF-1 degradation by inducing Mdm2. As a result, the dominant player in the network switches from HIF-1 to p53 upon severe hypoxia. In this process, HIF-1 first facilitates p53 accumulation by inducing PNUTS, and then p53 level further rises due to ATR-dependent stabilization and represses HIF-1 by competing for p300 from it. p53 accumulates progressively in response to hypoxia, which may imply the difficulty for p53 to be activated under hypoxia in contrast to the case in the DNA damage response [42]. Moreover, p53 only induces some target genes upon severe hypoxia and transcriptional repression becomes a significant way to induce apoptosis [92,93]. The coregulator CSB increases the complexity in the interplay between HIF-1 and p53, i.e., under moderate hypoxia, activated HIF-1 promotes its own activation by inducing CSB to dissociate p300 from p53 [94]. In contrast to the role of HIF-1 in promoting glycolysis, p53 primarily inhibits this process by inducing TP53-induced glycolysis and apoptosis regulator to repress fructose-2,6-bisphosphatase or repressing NF-κB-mediated upregulation of some glucose transcription factors [95,96,97]. Thus, the interaction between p53 and HIF-1 networks ultimately decides the cellular outcome in response to hypoxia.

It was reported that the interference of NF-κB signaling by inhibiting IKK leads to delayed peak timing and prolonged periods in p53 oscillations, suggesting a coupling between p53 and NF-κB signaling pathways [98]. The inhibition of NF-κB signaling affects p53 dynamics, depending on several interactions between these two pathways. First, they can all be activated by IR or inflammation via the ATM pathway. Deletion of IKK can amplify p53 signaling because of the presence of mutual inhibition of IKK and p53 [99]. Furthermore, both p53 and NF-κB can form a negative feedback loop with Mdm2, resulting in a mutual inhibition between p53 and NF-κB [100]. Indeed, similar to the interaction between p53 and HIF-1, there is a competition between p53 and NF-κB by snatching p300/CBP [101]. An exploration of a model depicting these complex interactions is likely to further explain the mechanism by which the presence of NF-κB pathway can accelerate the temporal evolution of p53 [98]. It remains an open topic to fully understand the interplay between these two pathways and explore some interesting questions, such as how cells reconcile p53-induced cell cycle arrest and apoptosis with the antiapoptotic role of NF-κB in response to stresses.

E2F-1 is a vital transcription factor that primarily regulates the cell cycle progression from G1 to S phase. In early G1, its transcriptional activity is repressed by pRB, while in late G1, E2F-1 is released and activated as pRB is phosphorylated by the Cyclin/CDKs (cyclin-dependent kinase) complex, driving the cells from G1 phase to S phase [102]. Due to its role in promoting proliferation, E2F-1 is exploited by cancers and it is overexpressed in many cancers, serving as an oncogene [103]. In contrast, in other cancers, such as prostate cancer, E2F-1 is considered to be a tumor suppressor by inducing apoptosis, thus establishing the dual function of E2F-1 in tumor progression. How to specifically amplify its cancer-inhibiting function while blocking its tumor-promoting function is a key question in targeting E2F-1 for cancer treatments. Some clues can be found in the exploration of the interaction between p53 and E2F-1. The investigation revealed that there is a negative feedback between these two transcription factors in which activated E2F-1 can enhance p53 signaling by upregulating ATM and ARF while activated p53 can inhibit E2F-1 by inducing p21 or miR-34 [104]. Maybe, we can utilize E2F-1 and DNA damage to amplify p53 signaling and make p53 dominant in p53-E2F-1 interactions in treating E2F-1 promoted cancers. More experiments and modeling-based efforts are called for to explore the mechanism of interaction between p53 and E2F-1 in response to DNA damage.

It is a challenge to understand the coactivation of p53 and TGF-β in the DNA damage response. Their interaction produces some paradoxical points: activated TGF-β can increase p53 expression by promoting the accumulation of ROS to amplify ATM activity [105]; at the same time it can repress p53 signal by upregulating Mdm2 through the activation of Smads [106]. These two actions form an incoherent feed-forward loop from TGF-β to p53. As a result, p53 may first rise due to the early positive regulation and gradually drop to low levels by the later repression from TGF-β pathway. Although the exact mechanism remains to be verified, the abundance of TGF-β could be a key factor in regulating p53 dynamics and affecting its tumor-suppressive functions. Moreover, the interplay between TGF-β and p53 expression levels suggests that TGF-β likely triggers p53 oscillations by activating p53 and triggering Mdm2 phosphorylation [107,108]. TGF-β plays a key role in promoting metastasis by inducing epithelial-to-mesenchymal transition (EMT) since TGF-β pathway can induce several EMT-related transcription factors including Snai and Zeb [109]. The mutual inhibition between these EMT-TFs and miRNAs (including miR-34 and miR-200) results in tristability in the levels of E-cadherin, whose loss is usually considered as a marker of EMT [110]. Strikingly, p53 inhibits tumor metastasis by upregulating inhibitors of EMT-TFs (e.g., miR-34) and thus competes with TGF-β in the control of EMT [111]. It is an urgent task to manipulate the interaction of p53 and TGF-β so as to simultaneously enhance TGF-β-mediated activation of p53 to trigger apoptosis and increase miR-34 expression to inhibit metastasis in cancer therapy. Cycle arrest is a major cell fate mediated by p53. Indeed, TGF-β can also regulate this important process by modulating the concentration of p21. In particular, Xing et al. proposed two potential mechanisms for the coupling of EMT to the cell cycle [112]. Therefore, it is an interesting topic to consider how the cell cycle is regulated when considering the activation of the p53 network by DNA damage to treat TGF-β-mediated metastatic tumors.

## 5. Conclusions

Over the last 20 years, the development in experimental techniques has advanced the measurement of p53 dynamics in individual cells. In general, p53 dynamics depend on the type of stimulus, cell, tissue, and species. In addition, the detection methods have also been found to impact the dynamics of p53 obtained by experiments [113]. Both experimental and modeling researchers are endeavoring to uncover how such dynamics arise and their impact on cell fate decisions. Integrating multiple experimental results is required to establish a functional network that responds to a specific stimulus. Network modeling may be helpful for gaining a global understanding of the whole process from encoding to decoding of p53 dynamics in cellular response to stresses. In different cell lines, p53 exhibits distinct dynamical sensitivity when exposed to a specific agent, such as the antitumor drug etoposide [6]. This finding suggests that we should select an appropriate antitumor agent according to the cell type when treating cancers. The modeling results revealed that the same network with variable interaction strengths could account for the different sensitivity in p53 dynamics in response to stresses [7]. Probing the mechanism in the conversion between different p53 dynamic modes is still a challenging topic in the field of p53 research. For example, fluctuations in p53 pulsing could induce the switching of p53 to sustained high expression while the underlying mechanism is still less understood [114]. Moreover, multiple layers of regulation including posttranslational modifications can regulate the dynamics of p53 and assist in identifying different target genes. Here, we focus on Mdm2/MdmX as the primary regulators of p53 activity and stability. However, in other cell types including neurons, p53 dynamics are modulated by other factors that regulate p53 stability, such as calpain [115,116]. From the perspective of killing cancer cells by increasing p53 expression, a closer inspection of the p53-Mdm2 loop would enhance the efficiency of cancer therapy. Comprehensively considering the mechanisms that promote the efficiency of p53 accumulation with other factors, such as the resulting side effects [88,117], may provide clues to cancer treatment.

## Figures and Tables

**Figure 1 cells-12-00490-f001:**
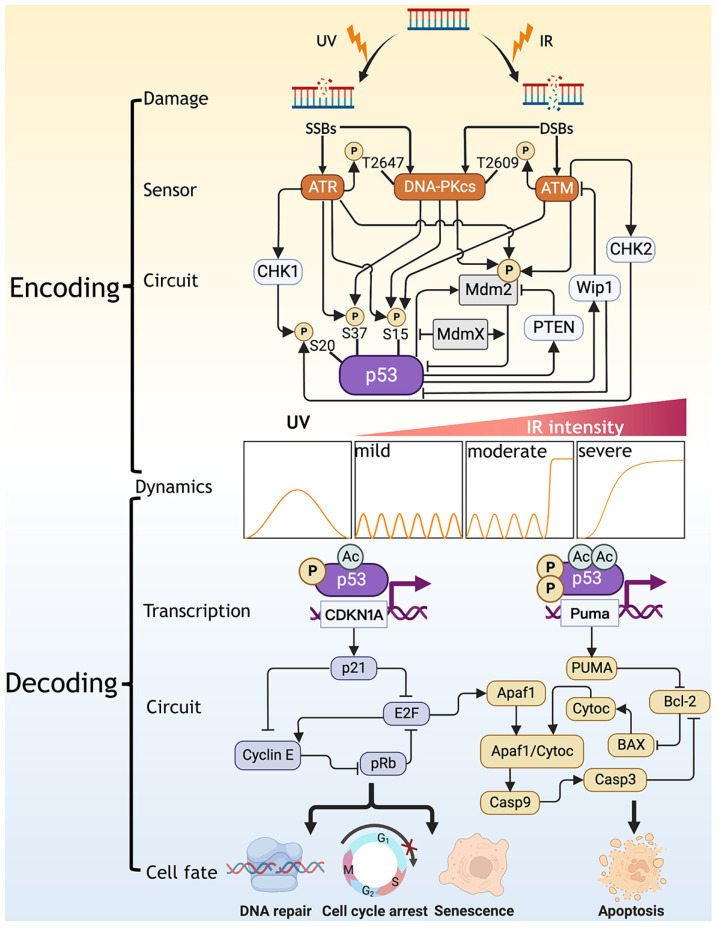
Schematic diagram of the network accomplishing the encoding and decoding of p53 dynamics. The upstream of the network includes DNA damage, sensor, and circuits involved in p53 regulation, contributing to the encoding of p53 dynamics. The downstream of the network contains the factors that decode p53 dynamics into specific cell fates. p53 dynamics act as a linker between the encoding and decoding process. There exist four types of p53 dynamics corresponding to the case in response to UV, IR (mild damage), IR (moderate to severe damage), and IR (extremely severe damage). Cytoc, Casp3 and Casp9 are the abbreviation forms of cytochrome c, caspase 3 and caspase 9, respectively. The arrow- and bar-headed lines indicate promotion and inhibition, respectively.

**Figure 2 cells-12-00490-f002:**
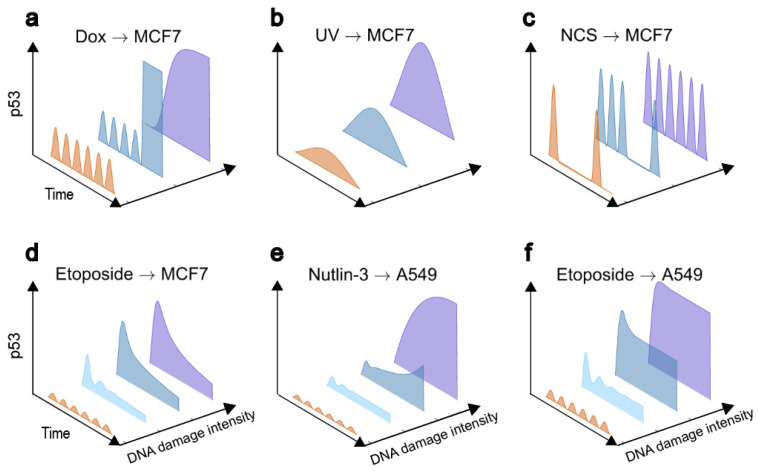
DNA damage intensity-dependent conversion of p53 dynamics. In MCF7 cells, Dox (**a**), UV (**b**), and NCS (**c**) induces different transition dynamics of p53 in patterns with two-phase (transient oscillations followed by rising to high levels), extended pulse, and two-phase (transient oscillations followed by spontaneous pulses), respectively. (**d**) In etoposide-treated MCF7 cells, the p53 transition dynamics consisted of two modes: first undergoing a damped oscillation and then a lower height pulse. (**e**,**f**) In A549 cells treated by Nutlin-3 (**e**) or Etoposide (**f**), p53 shows damped oscillations with different amplitudes.

**Figure 3 cells-12-00490-f003:**
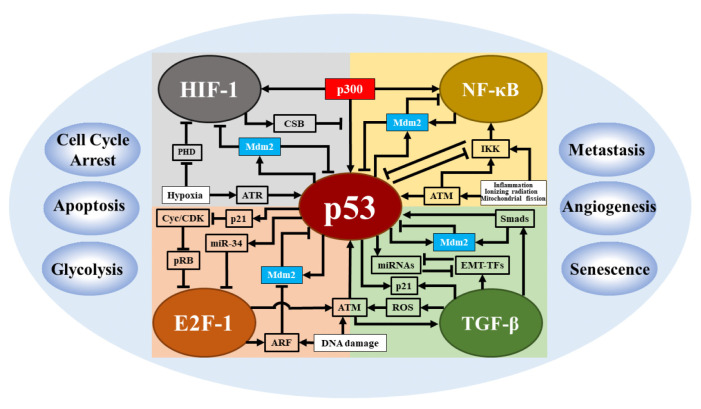
Coupling of p53 network with other signaling networks. (Top left) Interplay between p53 and HIF-1 in hypoxia: p53 and HIF-1 are coupled together through p300, CSB, and Mdm2, modulating processes including cell cycle arrest, angiogenesis, and apoptosis. (Top right) Coupling of p53 and NF-κB: The two transcription factors are interlinked via ATM, IKK, p300, and Mdm2 to regulate glycolysis and apoptosis. (Bottom left) Coordination of p53 and E2F-1: p53 can regulate E2F-1 through p21-Cyc (cyclin)/CDK (cyclin-dependent kinase)-pRB pathway and miRNAs, while E2F-1 affects p53 through ARF and MDM2, thereby modulating cell cycle arrest and apoptosis. (Bottom right) Reciprocal regulation between p53 and TGF-β: There is a positive feedback loop between ATM and TGF-β. TGF-β regulates the activity of p53 mainly through Smads. p53 and TGF-β compete to regulate EMT program in the process of metastasis. p53 and TGF-β cooperate to regulate cell cycle arrest. The arrow-headed lines indicate promotion, while the bar-headed lines indicate inhibition.

## Data Availability

No specific data in this review.

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
