# Peer review of "Encoding and Decoding of p53 Dynamics in Cellular Response to Stresses"

_cells, 2023, doi:10.3390/cells12030490_

Round 1

Reviewer 1 Report

In this review paper, Wang et al. provided an excellent and comprehensive review of the encoding and decoding of p53 dynamics in the cellular response to various stresses. The manuscript is well written, the structure is well designed, the complicated information from the publications is well organized, and the discussion is very insightful. I believe that the whole systems biology community has been looking forward to seeing the deep insights from this great review. Thus, I fully support the publication of this review in Cells after some minor changes.

1.     Page 4, it would be much easier for the audience if the cross-references of the specific panel of figure 2 were added when talking about each mode of p53 dynamics depending on the stimulus cell type and species.

2.     In Fig 2, it would be less misleading if the x z axis labels ‘p53’ and ‘DNA damage intensity’ were placed inside the box.

3.     The design of coupling of the p53 network with other signaling networks in Fig.3 is very nice. It would be great to specify the meaning of different arrows, especially the one toward p300.

4.     The discussion on the crosstalk between p53 and EMT signaling pathways is very insightful. It is noted that EMT is also coupled with cell cycle arrest. Two potential mechanisms were proposed (PMID: 30665206). Given that p53 is also a key regulator of cell cycle arrest, there could be coordination between p53 and EMT on the regulation of cell cycle arrest in addition to other cell fate transitions.

Author Response

1. Page 4, it would be much easier for the audience if the cross-references of the specific panel of figure 2 were added when talking about each mode of p53 dynamics depending on the stimulus cell type and species.

Reply: Thank you very much for this suggestion. We have added cross-references to Fig.2 in the section “2.2. Stimulus-, cell type- and species-dependent p53 dynamics”, on page 4, lines 128, 143, and 148, respectively. Likewise, given that our reference to Figure 1 is also inadequate, we have added the sentence " To give an overview of p53 response to DNA damage, we summarize the p53 signaling network including the damage types, sensors, upstream circuits, dynamics, transcription, and downstream circuits associated with different cell fates. " in the last paragraph of the section “Introduction” (line 65) and cited Fig.1 at the end of the sentence (line 67).

2. In Fig 2, it would be less misleading if the x z axis labels ‘p53’ and ‘DNA damage intensity’ were placed inside the box.

Reply: As suggested, we moved the x and z-axis labels "p53" and "DNA damage intensity" to the appropriate positions, and also removed the outer box that seemed unnecessary after the shifting of these two labels.

3. The design of coupling of the p53 network with other signaling networks in Fig.3 is very nice. It would be great to specify the meaning of different arrows, especially the one toward p300.

Reply: To make the figure easily understood by the readers, at the end of the caption of Figure 3, we have added the sentence "The arrow lines indicate promotion, while the bar-headed lines indicate inhibition." to clarify the meaning of each arrow.  

4. The discussion on the crosstalk between p53 and EMT signaling pathways is very insightful. It is noted that EMT is also coupled with cell cycle arrest. Two potential mechanisms were proposed (PMID: 30665206). Given that p53 is also a key regulator of cell cycle arrest, there could be coordination between p53 and EMT on the regulation of cell cycle arrest in addition to other cell fate transitions.

Reply: In Fig.3, we have added the regulation of p21 by p53 and TGF-βand illustrated this regulation in its caption by adding the sentence “p53 and TGF-β cooperate to regulate cell cycle arrest.”; also, we added the following sentences “Cycle arrest is a major cell fate mediated by p53. Indeed, TGF-β can also regulate this important process by modulating the concentration of p21. In particular, Xing et al. proposed two potential mechanisms for the coupling of EMT to the cell cycle [112]. Therefore, it is an interesting topic to consider how the cell cycle is regulated when considering the activation of the p53 network by DNA damage to treat TGF-β-mediated metastatic tumors.” at the end of the fourth paragraph on page 13 (Line 590). The reference PMID: 30665206 was cited as Ref. [112].

Reviewer 2 Report

The authors review the different studies investigating whether the p53-mediated cell fate decision can be predicted by measuring the oscillation pattern and intensity of the p53 expression level (p53 dynamic). It is an excellent review. It is exhaustive and it accurately cites original publications.  

Major points:

However, the authors focus on MDM2/MDMx as if they were the main regulators of p53 activities while p53 is degraded by other ubiquitin ligases and proteases such as calpain. The authors should make loud and clear to the readers that MDM2/MDMx are not the only regulators of p53 protein expression level. Importantly, the authors have included a paragraph describing the effect of post-translational modifications on p53  transcriptional activities highlighting that p53 expression level is NOT the only parameter defining the cell fate decision. It is an important paragraph as it emphasizes to the readers that the post-translational modifications of p53 define its subcellular localization, protein-protein interaction, and protein-DNA interactions and thus its tumor suppressive activities. The post-translational modifications are important to decipher the p53 code. 

The authors should add in figure1 the interplay between ATM, ATR, DNA-PK, MDM2/MDMx, and p53. The author should show the phosphorylation sites of ATM, ATR, and DNAPK on p53 protein as well as the phosphorylation site of ATM on DNAPK.

The authors should highlight that ATM phosphorylates MDM2 (Mayat et al, Gene dev (2001) 15(9):1067-77; Magnussen et al 2020 Nat. Comm11, 2094).

The authors should also show that ATR phosphorylates MDM2 (Shinozaki et al., 2004, Oncogene 22, 8870-8880 and that DNAPK phosphorylates MDM2 ( Mayo et al., Cancer res. 15, 57(22), 5013-16).

The interplay between TGFbeta and p53 expression level suggest that TGF-beta induce oscillation of p53 expression level probably by inducing the phosphorylation of MDM2 (Araki et al, JCI 2010; 120(1): 290–302)

Minor revision:

Line 366: please change “multiple transcriptional modifications of p53”  to “multiple post-translational modifications of p53”

Line 374: “8 types of post-translational modifications”, This is not accurate. There are more than 8 types of post-translational modifications of p53. Please modify

Line 583: the authors should state that the type of detection methods impacts the p53 dynamics in  addition to the type of stimuli, cells, tissues, and species.

Author Response

Major points:

1. However, the authors focus on MDM2/MDMx as if they were the main regulators of p53 activities while p53 is degraded by other ubiquitin ligases and proteases such as calpain. The authors should make loud and clear to the readers that MDM2/MDMx are not the only regulators of p53 protein expression level. Importantly, the authors have included a paragraph describing the effect of post-translational modifications on p53 transcriptional activities highlighting that p53 expression level is NOT the only parameter defining the cell fate decision. It is an important paragraph as it emphasizes to the readers that the post-translational modifications of p53 define its subcellular localization, protein-protein interaction, and protein-DNA interactions and thus its tumor suppressive activities. The post-translational modifications are important to decipher the p53 code. 

Reply: To indicate that Mdm2 and MdmX are not the only major regulators of p53 in the presence of DNA damage, we have added the following sentencesHere, we focus on Mdm2/MdmX as the primary regulators of p53 activity and stability. However, in other cell types including neurons, p53 dynamics are modulated by other factors that regulate p53 stability, such as calpain.to the section Conclusions (line 615).

2. The authors should add in the figure1 the interplay between ATM, ATR, DNA-PK, MDM2/MDMx, and p53. The author should show the phosphorylation sites of ATM, ATR, and DNAPK on p53 protein as well as the phosphorylation site of ATM on DNAPK.

Reply: To emphasize the importance of post-translational modifications in deciphering the p53 code, we have made a comprehensive revision of Figure 1: (1) we added the interplay between ATM, ATR, DNA-PK, MDM2/MDMx, and p53; (2) we showed the phosphorylation sites of ATM, ATR, and DNA-PK on p53 protein as well as the phosphorylation site of ATM on DNA-PK; (3) we showed MDM2 phosphorylation by ATM, ATR, and DNA-PK.

3. The authors should highlight that ATM phosphorylates MDM2 (Mayat et al, Gene dev (2001) 15(9):1067-77; Magnussen et al 2020 Nat. Comm11, 2094).

Reply: To highlight ATM-dependent MDM2 phosphorylation, we have added the sentence " Moreover,ATM-dependent phosphorylation of Mdm2 can accelerate its degradation, thereby promoting p53 activation indirectly;" in page 3, line 100. The mentioned references have been cited in the above sentence.   

4. The authors should also show that ATR phosphorylates MDM2 (Shinozaki et al., 2004, Oncogene 22, 8870-8880 and that DNAPK phosphorylates MDM2 ( Mayo et al., Cancer res. 15, 57(22), 5013-16).

Reply: We showed the role of ATR in Mdm2 phosphorylation by adding the sentence “while ATR-dependent Mdm2 phosphorylation only inhibits the activity of Mdm2 for degrading p53.” We indicated that DNA-PK can phosphorylates both p53 and Mdm2 in the sentence “DNA-PKcs can be activated by both DSBs and SSBs, stabilizing p53 by phosphorylating both p53 and Mdm2.” In page 3, line 102. The mentioned references have been cited in the above sentence.   

5. The interplay between TGFbeta and p53 expression level suggest that TGF-beta induce oscillation of p53 expression level probably by inducing the phosphorylation of MDM2 (Araki et al, JCI 2010; 120(1): 290–302)

Reply: Following your suggestion, we have added the sentence “Moreover, the interplay between TGF-β and p53 expression levels suggests that TGF-β likely triggers p53 oscillations by activating p53 and triggering Mdm2 phosphorylation.” (Line 579).

Minor revision:

1. Line 366: please change “multiple transcriptional modifications of p53” to “multiple post-translational modifications of p53”

Reply:In the revised version, we have changed “transcriptional” to “post-translational” in line 372.

2. Line 374: “8 types of post-translational modifications”, This is not accurate. There are more than 8 types of post-translational modifications of p53. Please modify

Reply:We have replaced the related sentence in the previous version with the sentences “The main types of post-translational modifications on p53 include SUMOylation, neddylation, phosphorylation, ……modifications of p53. [72,73]. These modifications …… the post-translational modifications of p53. [72,73].” in line 380.

3. Line 583: the authors should state that the type of detection methods impacts the p53 dynamics in addition to the type of stimuli, cells, tissues, and species.

Reply: We have added the sentence “In addition, the detection methods have also been found to impact the dynamics of p53 observed by experiments.” in line 599.

Reviewer 3 Report

Studies focusing on mathematical models of p53 dynamics in response to DNA damage have been reviewed by Wang et al. In this review study, Wang et al highlighted the important role of p53 in cell fate decisions, which had been captured or predicted by mathematical models.

The article is well written.

The references are well up to date.

This review article is well organized.

The figures are fine.

I have two suggestions for improving their manuscript.

Concerning Figures 1 and 3, the authors should clarify in the captions of the figures what is meant by the hammer arrows (which are inhibitions) and the arrows (which are activations).

Concerning Figure 1, at the end/bottom of the figure, we can see "cell fate", in which the authors show a sequence of outcomes for p53,  which is based on the different phosphorylation states of p53 and resulting in, cell cycle arrest, DNA repair, apoptosis, and senescence in DDR. Therefore, the sequence of the p53 outcomes is as follows DNA repair, cell cycle arrest, senescence, and apoptosis. More detail can be found on PMID: 25483060.

Overall excellent work.

Author Response

1.Concerning Figures 1 and 3, the authors should clarify in the captions of the figures what is meant by the hammer arrows (which are inhibitions) and the arrows (which are activations).

Reply: As suggested, we separately added the sentences “The arrow- and bar-headed lines indicate promotion and inhibition, respectively.” and “The arrow-headed lines indicate promotion, while the bar-headed lines indicate inhibition.” in the captains of Figures 1 and 3, respectively.

2. Concerning Figure 1, at the end/bottom of the figure, we can see "cell fate", in which the authors show a sequence of outcomes for p53, which is based on the different phosphorylation states of p53 and resulting in, cell cycle arrest, DNA repair, apoptosis, and senescence in DDR. Therefore, the sequence of the p53 outcomes is as follows DNA repair, cell cycle arrest, senescence, and apoptosis. More detail can be found on PMID: 25483060.

Reply: We have rearranged the four cell fates in Figure 1 in the order of "DNA repair, cell cycle arrest, senescence, and apoptosis".